# Regional Densities of Cooperation: Are There Measurable Effects on Regional Development?

**Christian Diller [1],\*, Guido Nischwitz [2], Martin Kohl [1] and Patrick Chojnowski [2]**

[1]   Institute of Geography, University of Giessen, D-35390 Giessen, Germany; m.kohl@umwelt-campus.de
[2]   iaw—Institute Labour and the Economy Bremen, University of Bremen, D-28359 Bremen, Germany; gnischwitz@uni-bremen.de (G.N.); s_patrick.chojnowski@uni-bremen.de (P.C.)
\*   Correspondence: christian.diller@geogr.uni-giessen.de

**Abstract:** Almost three decades ago, a paradigm change in funding policies for rural regions became effective in Europe and Germany, involving a move towards cooperative, actor-oriented regional development. However, little research has been published on the extent to which funding approaches intended to activate cooperation have led to regional-economic effects in the regions. This paper presents a countrywide statistical evaluation of the link between the deployment of funding programmes and established regional development indicators. The investigation is based on the analysis of 27 funding programmes, pilot projects and competitions from five policy fields, covering the period from 1991 to 2016. Its analyses are founded on the largest database of regional-development programmes implemented in Germany and the first attempt to detect cumulative effects of a large number of programmes over a long period. Further research in this direction should first gather detailed information on the scope of funding programmes in the regions.

**Keywords:** regional governance; regional cooperation; networks; measurable effects; regional economy; state funding; Germany

---

## 1. Introduction

### 1.1. Impacts of Regional Funding Policies

Almost three decades ago a paradigm change in regional funding policies took effect in Europe and Germany, with a focus on 'endogenous, integrated regional development'. A bottom-up approach was applied with the intention of activating a broad spectrum of regionally generated potential. State interventions in the EU and Germany aim to initiate and actively guide the creation of new spatial structures and relations with the objective of enhancing activities on the regional level and strengthening regional capacities [1]. In addition to classical funding programmes, innovation-promoting instruments such as pilot projects and competitions are increasingly utilised. The overall picture of regional development policies in the multi-level political system is very heterogeneous and differentiated.

The theoretical foundation underlying this paradigm change is provided by concepts from 'relational economic geography' [2] that, in contrast to classical regional-economic approaches, focus more on the institutional context of a space and interactions between actors. The theoretical stringency of these concepts and the robustness of their empirical evidence have been criticised within economic geography [3] (p. 243). On the other hand, research within political science has had little success in providing solid empirical data to demonstrate the impact of regional management structures and institutions on regional development.

## 1.2. State of Research

The research field of (regional) governance has attracted a great deal of attention in the last 15 years. Basic research on governance concentrates on the structural characteristics, mechanisms [4] and patterns of development [5] of regional management structures, with a stronger focus on the question of processes of learning [6].

First, an overview of the national funding programmes in Germany is given: most nationwide studies on GRW promotion in Germany conclude that it has beneficial economic effects on the supported regions and allows them to draw level with other regions: [7,8] for the German labour market regions between 1994 and 2006 [9,10], for the German labour market regions between 1999 and 2006 and between 1999 and 2008; [11] in 271 German labour market regions between 1992 and 2003. However, other research on this programme showed negative results: regional effects and spatial spillover effects of investment support were examined by Eckey/Kosfeld [12] on the basis of German labour market regions in the funding period of 2000 to 2002. It was found that the regional crowding-out effects were very high. Thus, about 96 percent of regional funding was used without success. The evaluation of the IWF Halle [13] analysed the promotion of business support infrastructure within the GRW in Saxon municipalities during the period from 2000 and 2007. It was shown that subsidies tended to be allocated to those municipalities with a high structural and financial strength and in general had a rather restricted positive impact on economic development. The GRW funding in Thuringia in the years from 2004 to 2010 was evaluated in an analysis by the GEFRA [14]. The evaluation demonstrated that no distinction could be made between a focus of GRW promotion on the state's slow-growing or fast-growing regions. Neither could it be shown that a high level of GRW funding is linked to a higher than average productivity growth.

A more detailed look will now be given at international evaluation programmes for EU regional development. A longstanding objective of EU regional policy is to reduce socioeconomic disparities between the EU member states and their regions [15]. However, the strategies for achieving this goal changed between the programme periods: Since the early 2000s elements such as growth, competition and the enhancement of strengths have received more attention [16]. The first group of studies examines all regions of the EU member states on several spatial levels (NUTS 2 or/and NUTS 3).

By using a continuous regression model Becker et al. [17] examined the impact of the EU structural funds on regional performance. The programme periods considered in this evaluation are 1989–1993, 1994–1999 and 2000–2006 on the NUTS 2 level. The study found a positive correlation between Objective One promotion and per capita GDP growth in the supported regions during the programming period. However, it was not possible to find any verifiable effects on employment. The first reason for this is that the support only affects the size and structural aspects of the investments. Moreover, the impact on job creation can only be measured in the long term. Another study by Becker et al. [17,18] analyses to what extent the intensity of EU regional policy support in the programming periods 1994–1999 and 2000–2006 affects regions at NUTS 3 level. In order to assess the impact of various financing intensities on per capita growth, an estimate of GDP is applied. The key finding is that the most efficient level of funding can be found at 0.4 percent of a region's GDP. On the other hand, regions with a promotion level of more than 1.3 percent of their GDP can dispense with subsidies while still maintaining per capita income growth. Thus, 1.3 percent of a region's GDP represents the maximum level of assistance. Breidenbach et al. [19] use panel data for an assessment of the effects of Objective One promotion on the goal of income convergence in the 127 NUTS-2 regions of the EU-15, covering the time frame from 1997 to 2007. The authors conclude that Objective One aid appears to have either no or an even negative impact on regional growth in the regions studied. The reason for the overall negative impact is mainly attributable to negative spillover effects on adjacent areas. In contrast, Gagliardi/Percoco [20] in their study on the impact of EU funding on regional development (NUTS 2 and NUTS 3-level) for the period 2000 to 2006 came to the finding that the subsidies basically had positive effects on the development of the regions. Nevertheless, the effects were more positive for regions relatively close to agglomerations than for the peripheral regions. Thus, the result of the EU promotion was a

widening of the gap between the regions of this type that received support. A very similar result was identified by Crescenzi/Giua [21] in their research on almost all European regions between 2004 and 2013. A certain positive influence of GDP was only evident in the most advanced and better equipped areas, in other words in the rural areas of the 'core' of the EU and not in the most disadvantaged and peripheral regions.

The second group of studies examines the impacts of EU regional funding on individual countries: In their analysis for 17 Spanish regions in the years 1989 to 2010, Faiña et al. [22] found that the EU funding for transport infrastructure at least contributed to the decline in economic growth during this same period. A further analysis for Spanish regions [23] also examined a positive effect of EU funding on their economic development. Similar results were found in a study for Polish regions [24]. In the UK Di Cataldo/Monastiriotis [25] found positive impacts for all funded regions during the period from 1994 to 2013, with more positive effects in the stronger regions receiving grants. Only one study came to conclusion that EU funding had no impact: For Hungary, Bakucs et al. [26] found that socioeconomic effects of the EU funding were not measurable, using a counterfactual analysis for the period 2002 to 2008.

In addition, some examples of evaluation of national programmes in EU countries can be added: Acceturo and De Blasio [18] evaluated the Patti Territoriali development programme in Italy between 1996 and 2001. Methodologically, this work is based on a counterfactual analysis. The authors concluded that the programme showed only minor impact both in terms of increasing employment opportunities, as well as the number of enterprises. It can therefore be considered highly ineffective. A possible reason for this could be the restricted public expenditure of 50 million euros per programme. The evaluation of De Castris/Pellegrini [19] focuses on the spatial effects of capital subsidies. The study examines regions in southern Italy using a spatial autoregressive model for the years 1996 to 2001. Overall, the support influenced the beneficiary companies positively. However, a displacement effect in the immediate and surrounding regions is evident. By attracting new investment to the areas through political intervention, these measures generate new employment opportunities.

Finally, there are a few studies that investigate the cumulative effects of several different regional development programmes, while at least taking an integrated view. Coppolla et al. [27] examined the impact of both EU structural support and national regional development programmes on economic growth between 1994 and 2013 for 20 Italian regions. Significant effects could be measured for EU funding, but not for the national programmes. In contrast, Psycharis et al. [28], with their analysis of EU and national funding in Greece for the period 2000 to 2014, concluded that the national programmes had significant impacts, but not the EU programmes. Crescenzi and Giua [21] came to a further conclusion in their study for almost all European regions: the linkage of space-oriented EU programmes with sectoral grants led to positive interdependencies [21].

Alecke et al. [7] evaluated the effects of regional policy on economic growth by examining the German national GRW funding and the ERDF funding of the EU. Its key finding was that regional policy yielded a significant positive impact on a region's labour productivity and convergence rate. In regions that are well below the steady-state income, these effects are the strongest. Furthermore, they also grow with the amount of funding allocated to adjacent regions. In addition, the attractiveness of the whole of Germany is increased by positive spatial spillover effects.

In summary, the impact analyses came to very different results concerning the impact of the regional development funding.

For EU programmes, most studies for all countries or individual countries for different programming periods have found that funding has had a positive impact on the economic development of the regions. However, the results are unsatisfactory when compared to the policy objectives: the highest impact was found for those regions that were among the strongest of the regions supported. Thus, the undesirable result of EU funding is in some cases a reduction in development differences between the strong regions without funding and the group of relatively strong regions among the regions with subsidies. On the other hand, the regions with the lowest growth rates are more peripheralised

in their development rates despite the financial support. The few studies that examined several programmes simultaneously came to very different findings. In some cases, EU programmes have more impacts than national programmes while in other cases the relationship is reversed. One study recognised positive interdependencies between programmes.

The differences in the measured effects of different programmes lead to the assumption that the national and regional contexts of programme implementation have an influence on the effectiveness and efficiency of the programmes [29–33]. For a deeper analysis of the reasons, it is necessary to clarify the exposure pathways of the funding programmes [34].

Basic research has thus far paid little attention to assessing the effects of 'regional governance' on the system of regional economic development or interactions between the two systems, with few notable exceptions [6,35,36]. This task has rather been delegated to evaluation research [37], a field that has progressed significantly in the last 20 years, parallel to the increased differentiation of funding programmes particularly on the EU level [38]. Thus, for the majority of EU programmes, particularly those relevant to structural policy (see [39]), there is a differentiated system of ex-ante, intermediate and ex-post evaluations, largely of high quality. The evaluations, however, are either mostly designed to monitor the implementation of programmes, concepts and projects or to assess the directly attributable effects of initiatives on the micro-level. There is very little research into the medium- and long-term impacts, especially on the development of rural areas [40] (p. 76).

There have been individual programme-related evaluations that have detected direct effects on employment. For instance, the countrywide competition REGIONEN AKTIV (2002–2007) 'REGIONEN AKTIV – Land gestaltet Zukunft' was a pilot project by the Federal Ministry of Food, Agriculture and Consumer Protection (BMELV) used to test new approaches for the funding and development of rural areas created or retained almost 1050 jobs in the 18 winning regions. This involved funding of EUR 56.30 million for about 1350 projects, leading to a further EUR 38.9 million being generated as public and private co-financing [41] (p. 9). Another example is the differentiated evaluation of the EU LEADER programme undertaken by Geissendörfer [42], in line with EU requirements, where result and impact indicators were operationalised. Overall, this evaluation judges the programme to have had positive effects. LEADER II. LEADER II was an EU initiative to support structurally weak rural areas, sponsored between 1994 and 1999 by the Directorates General for Regional Policy and Agriculture under the Structural Funds created a net total of just under 1600 jobs in Germany between 1994 and 1999 [42] (p. 170). However, such evaluations do not assess the indirect and lasting impact of approaches to regional cooperation on the development of the region as a whole, in particular they do not compare the overall development of assisted regions with that of regions receiving no funding [42] (p. 285).

The research undertaken by Panebianco [35] is of particular note in this context. This represented the first attempt to use aggregated statistical correlation analyses to assess on a broad, in some cases comprehensive, basis the impact of 'good regional governance' on the economic development of regions, with employment trends being used to measure the latter. Panebianco's most interesting finding is derived from numerous sub-investigations of the long-term development of employment in regions with explicit network approaches to cooperation compared to that of regions where cooperation was less developed. It cannot be proved that employment trends are affected by the dedicated regional structuring of management systems, the regional bundling of resources or the existence of different forms of regional networks. Nonetheless, other 'good-governance' variables were seen to correlate with positive economic development: efficient basic political and administrative structures, the actions of public actors and civic engagement; these are, however, factors that are located on the municipal and not on the regional level.

In 2014, Diller, Nischwitz and Kreutz [36] published the first countrywide statistical assessment of the link between the deployment of cooperative regional development programmes on the one hand, and established regional development indicators on the other hand. This research differed from other investigations in that it simultaneously considered a number of funding programmes from different

policy fields over a period of 15 years (1995–2009). The results of the analysis supported the findings of Panebianco [35]. They similarly provided no evidence that regions where cooperative approaches were promoted experienced more positive development than other regions. No long-term impact on regional development could be detected.

The following assessment takes up the work of Diller, Nischwitz and Kreutz [36] and develops it further. Firstly, a significantly larger number of funding programmes and programme regions are considered. Secondly, the analysis period is extended, and the methodology refined. Overall, 27 funding programmes, pilot projects and competitions (the following discussion does not distinguish between funding programmes, competitions and pilot projects), and almost 1500 programme regions are identified and analysed over a period of 25 years (1991–2016). These are correlated with selected regional development indicators, population trends, and the development of gross domestic product (GDP) (1995–2014).

## 2. Materials and Methods

The basis for the analysis was a database compiled and updated by the Institute for Labour and the Economy of Bremen University and the Department of Geography of the University of Giessen. The regional database was produced in the context of the German Research Foundation project 'Entwicklung eines Modells zur Analyse von Lebenszyklen regionaler Kooperationen in Multilevel-Regional-Governance' ('Development of a model for the analysis of the lifecycles of regional cooperations in multilevel-regional-governance' (Project number: GZ: DI 1641/9-1). The programmes considered can be assigned to five policy fields:

- Rural development policy;
- Spatial planning and regional development;
- Regional economic policy;
- Environment and nature protection;
- Research and education.

Figure 1 shows the examination steps of the whole research project. Only the first steps are relevant to this article; the other steps with their results are documented in other papers.

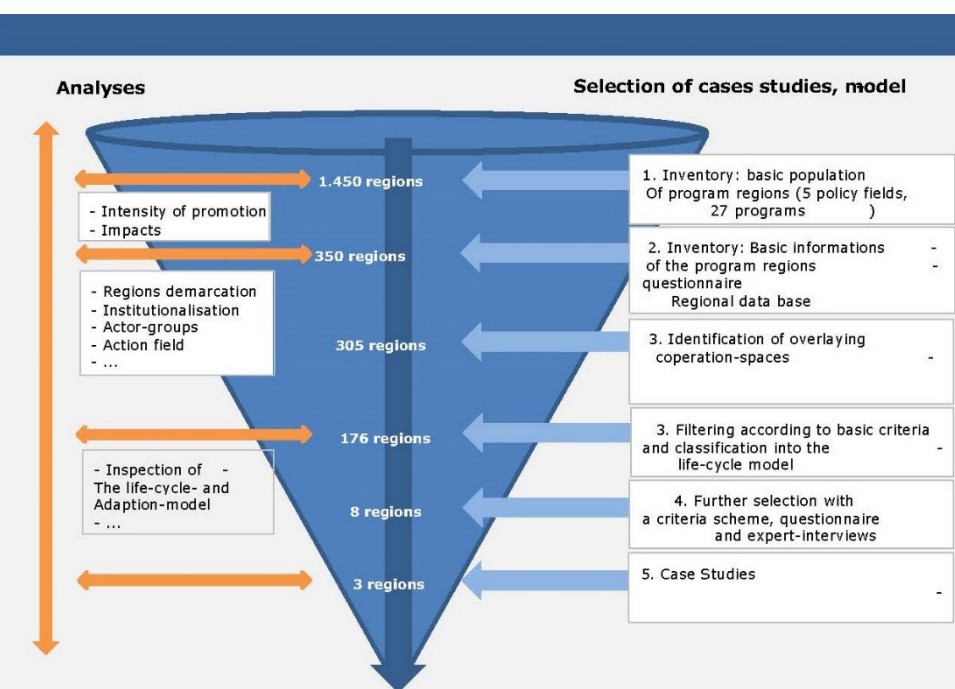

**Figure 1.** Steps of the empirical analysis.

When selecting programmes for the investigation a medium scale was chosen. This refers to a spatial level above that of a municipality and below that of a federal state. There is a conscious emphasis on programmes that focus on rural regions on medium scale, which, in size, is likewise to the German districts. This scale corresponds to the EU Nuts 3 level, and has already been used in other studies on the impact of regional programmes for Germany (see Section 1). It is the smallest scale on which the relevant dependent variables of socioeconomic development are available in official statistics.

The addresses of the funded regions were taken from the programme reports. The first information on the regions was also taken from the reports. Additional information was obtained through a questionnaire sent to the regions, telephone calls and e-mails. Figure 2 shows an overview of the programmes considered and their funding periods.

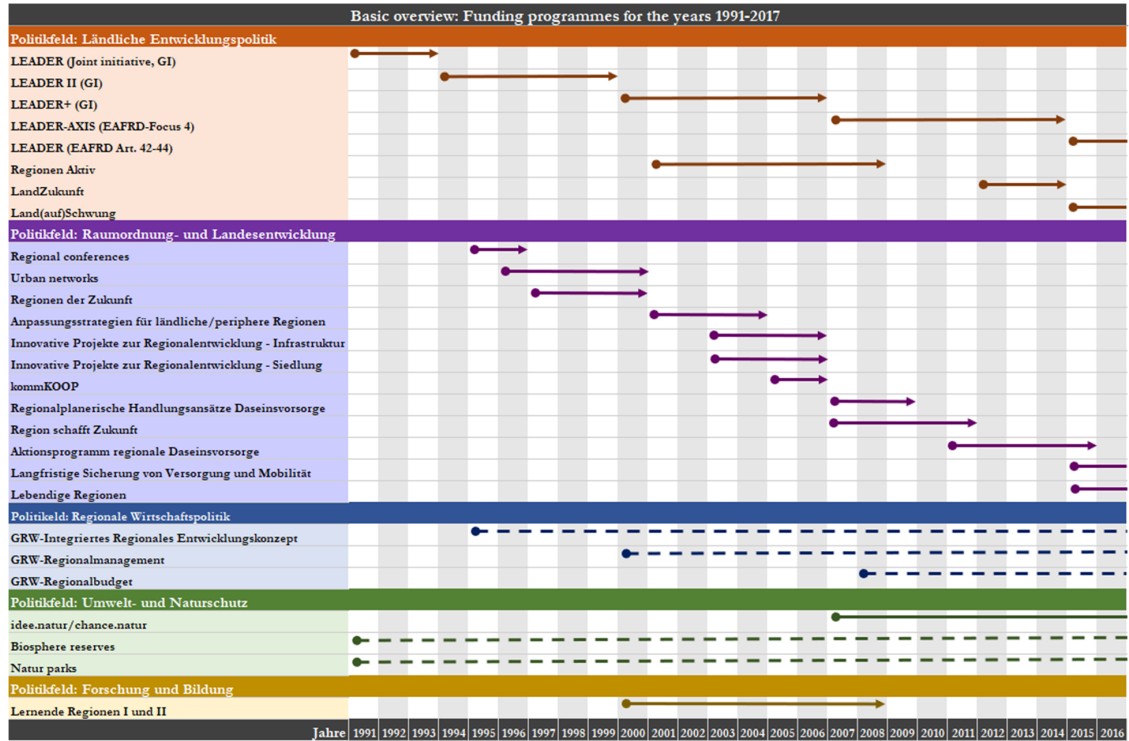

**Figure 2.** Basic overview of the programmes, dotted lines: varied starting- and end-points of funding programmes. Source: authors' compilation (Department of Geography Giessen, Institute for Labour and the Economy Bremen).

In an initial step the programme regions corresponding to the programmes analysed were first assigned to districts (Kreis) or towns with district status (Kreisfreie Stadt) in Germany. The districts (Kreise) and towns with district status (Kreisfreie Städte) are the highest level of the municipal body. At present, there exist 294 Kreise (with integrated core cities) and 107 kreisfreie Städte (separate core cities. It was not possible to take into account the differentiated and specific spatial form of the regions, which did and do not always exactly correspond with administrative district boundaries. Therefore, the analysis evaluates whether or not a programme is deployed in a district or town with district status. The number of programmes deployed in a district or town with district status was assessed. Finally, the method is used to count how many programmes have been implemented in each district. The implementation of each programme was only recorded once. If, for instance, several Local Action Groups (LAGs) of the LEADER programme existed in one district then they were only counted once so as to ensure comparability. A total of 1465 programme regions were finally entered into the database. On average, there are thus 3.6 programme-regions in a district or district town.

For the statistical analysis, demographic and economic data from the Federal Statistical Office (Statistisches Bundesamt) were added.

## 3. Results

The funding programmes investigated aim to initiate regional cooperation that should provide important impulses for successful and dynamic regional development. Central topics and fields of action include: the creation of equivalent living conditions, the stimulation of processes of cooperation, public services and demographic change, the creation of new jobs, the improvement of living conditions and the reduction of differences between successful and less successful regions. In contrast to Panebianco's [20] research, which considered employment trends, this investigation uses gross domestic product (GDP) per capita, the indicator of economic strength typically utilised in economic development studies, and population trends, the indicator that best represents the success of broadly based regional development policies intended to improve quality of life.

It should be noted that the following observations refer only to the programmes investigated and not to all of the regional initiatives funded in Germany.

### 3.1. Number of Programmes Deployed in the Districts and Towns with District Status

Figure 3 shows the distribution of assisted districts and towns with district status over the individual policy fields, programmes and groups of programmes. Over 70% of all districts and towns with district status (=401) participate in programmes in the policy field 'Rural Development' (orange bars), which include both a strategic (concepts, networking, management) and an investment (small scale infrastructure) focus. In the spatial planning field (purple bars) there are a notable number of individual programmes, although these provide funding to only relatively few districts and towns with district status.

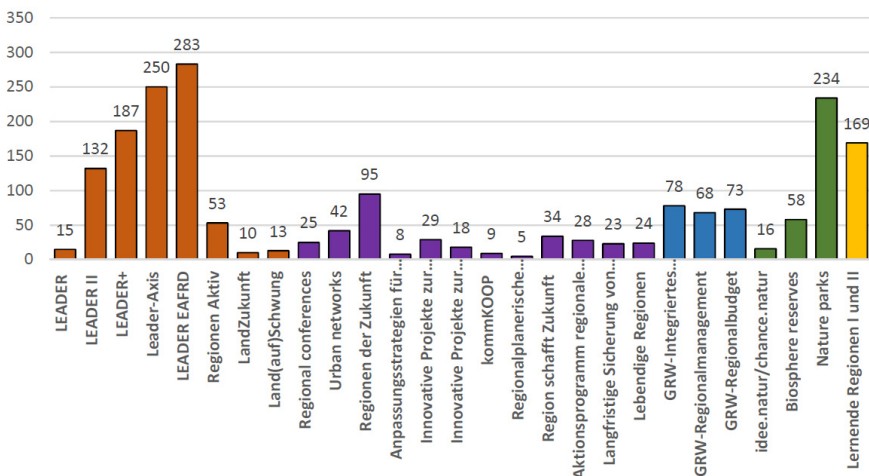

**Figure 3.** Number of assisted districts and towns with district status in the individual programmes (absolute numbers). *x*-axis: programmes: rural development policy (orange); spatial planning and regional development (purple); regional economic policy (blue); environment and nature protection (green); research and education (yellow); *y*-axis: number of assisted districts and towns with district status. Source: authors' compilation (Department of Geography Giessen, Institute for Labour and the Economy Bremen).

Figure 4 shows the number of funding programmes deployed (*x*-axis) in relation to the number of districts and towns with district status in which the respective number of programmes was deployed (*y*-axis). The mean number of funding programmes was five. Only 11% of the districts and towns with district status deployed less than two funding programmes between 1991 and 2016 (44 districts). Only 8.5% utilised ten or more funding programmes (34 districts). Examples of districts with more than 12 programmes are found in eastern Germany: Oderspreewald-Lausitz, Elbe-Elster and Spree-Neisse. Heading the field with 16 programmes is the district of Vorpommern-Greifswald.

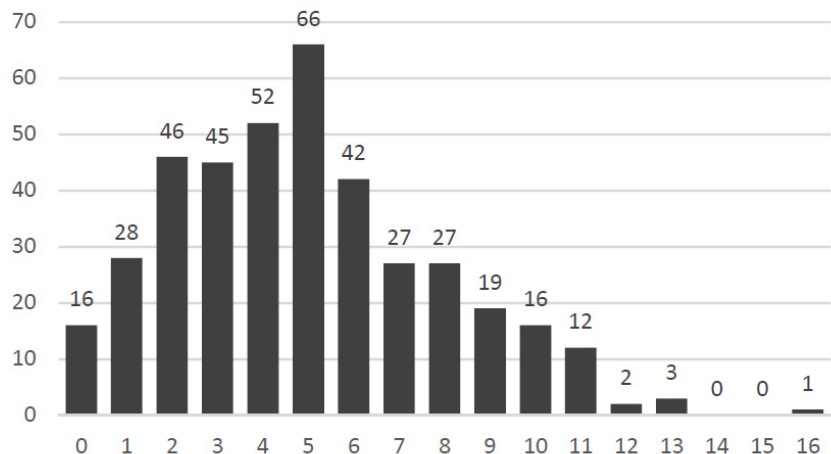

**Figure 4.** Number of funding programmes deployed and number of German districts and towns with district status in which the respective number of programmes was deployed (absolute numbers). *x*-axis: number of funding programmes deployed; *y*-axis: number of districts and towns with district status. Source: authors' compilation (Department of Geography Giessen, Institute for Labour and the Economy Bremen).

Figure 5 shows the geographical distribution. The district reforms implemented after 1990 in eastern Germany are taken into consideration (As the spatial extent of the district of Göttingen, newly amalgamated at the end of 2016, is equivalent to the form of the two former districts of Göttingen and Osterode am Harz, and it is included in the analysis as such).

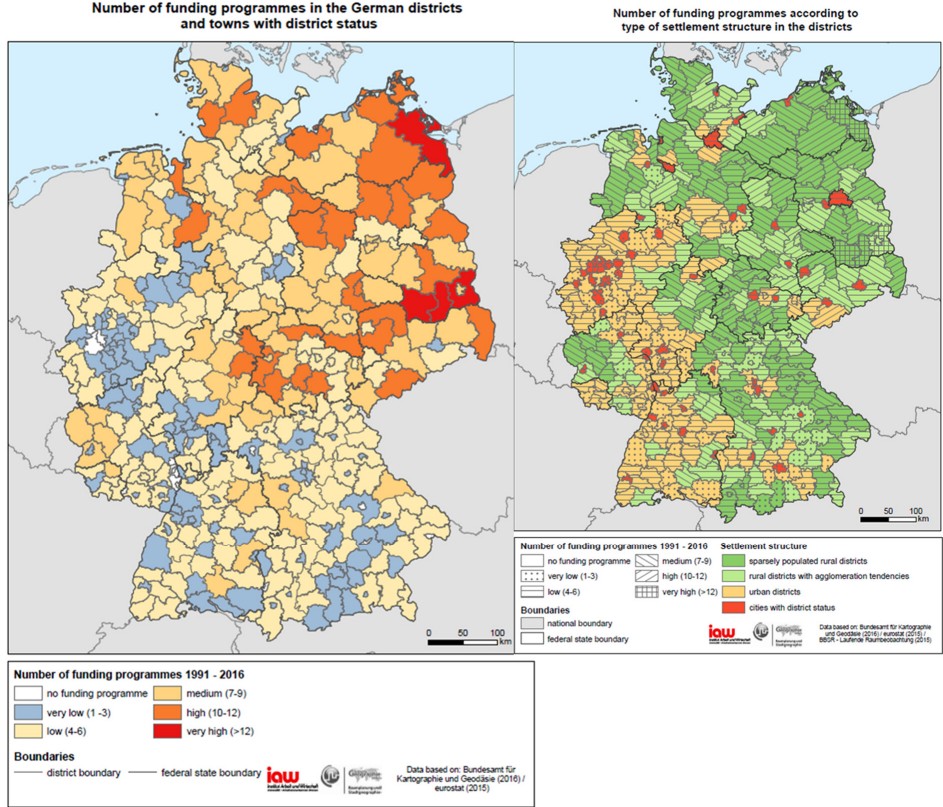

**Figure 5.** Number of funding programmes in the German districts and towns with district status (**left**) and number of funding programmes according to type of settlement structure in the districts (**right**). Source: authors' compilation (Department of Geography Giessen, Institute for Labour and the Economy Bremen).

Turning to a consideration of the intensity of funding programmes overall, it is possible to discern a slight discrepancy between north-east and south-west Germany in terms of the number of programmes deployed in districts and towns with district status between 1991 and 2016 (see Table 1). In the federal states of eastern Germany, Lower Saxony and Schleswig-Holstein (including the city states of Hamburg, Bremen and Berlin) the intensity of deployment of funding programmes is higher (n = 141, Ø = 7.26) than in the other states in western and southern Germany (n = 261, Ø = 3.71). A comparison with the mean of all districts and towns with district status (Ø = 4.95) confirms the relatively high intensity of funding programmes in the northern and eastern states. This is partly due to the choice of programmes considered in the analysis and the limitations set on the areas eligible for funding. Many programmes, such as LEADER or the MORO pilot project 'Aktionsprogramm regionale Daseinsvorsorge' ('Action programme regional public services') target structurally weak and disadvantaged regions, particularly those in rural areas. Nevertheless, not all rural regions in Germany lack economic development, as examples in Bavaria show.

**Table 1.** Number of programmes in the German districts and towns with district status.

| Intensity of Funding Programmes | n = 402 | Examples |
|---|---|---|
| No funding programme (0) | 16 | Rhine-District Neuss; towns like Düsseldorf, Würzburg and Regensburg |
| Very low (1–3) | 119 | Districts: Oldenburg, Viersen, Garmisch-Partenkirchen; towns like Dresden, Lübeck and Augsburg |
| Low (4–6) | 160 | Districts: Stormarn, Havelland, Kronach |
| Average (7–9) | 73 | Districts: Rostock, Plön, Schwäbisch-Hall |
| High (10–12) | 30 | Districts: Rendsburg-Eckernförde, Werra-Meißner-Kreis, Stendal |
| Very high (over 12) | 4 | Districts: Vorpommern, Greifswald, Oderspreewald-Lausitz, Elbe-Elster, Spree-Neiße |

Source: authors' compilation (Department of Geography Giessen, Institute for Labour and the Economy Bremen).

Focusing particularly on those districts and towns with district status with a high or very high number of funding programmes (see Table 1, left), it is revealed that only eight of these 34 territorial entities (ca. 24%) are not in eastern Germany. Only two of these districts (Hersfeld-Rotenburg and Werra-Meissner-Kreis) are not in the federal states found in the area of eastern and northern Germany described above.

Figure 5 (right) displays the number of funding programmes in the German districts and towns with district status according to type of settlement structure in the districts. The categorisation is based on the delimitations of the Federal Institute for Research on Building, Urban Affairs and Spatial Development, which use three characteristics of settlement structure: the proportion of the population in cities and medium-sized towns, the population density of the districts, and the population density of the districts excluding the cities and medium-sized towns.

If all the sparsely populated rural districts and rural districts with agglomeration tendencies (n = 199) are considered, it can be seen that about 16% (31 districts and towns with district status) deploy a high to very high number of funding programmes. Although this appears to be a low proportion, of the districts and towns with district status that are categorised as having an urban settlement structure (n = 137) there are only three that deploy a high to very high number of funding programmes. This represents a much lower proportion of only about two percent.

Focusing on only the districts and towns with district status that deploy a high to very high number of funding programmes (n = 34), it can be seen that about 91% are categorised as 'sparsely populated rural districts' or 'rural districts with agglomeration tendencies'. It can thus be concluded that the deployment of regional development programmes is correlated with the degree of rurality

of a district or town with district status. This finding is insofar not surprising: it shows that the programmes were implemented in regions in need of regional development impulses.

*3.2. Number of Programmes Deployed over Time (1991–2016)*

Focusing more closely on deployment over time (1991–2016) of the programmes included in the analysis, it becomes obvious that the number of programmes utilised in the districts and towns with district status has risen slightly. In 1991, only 197 districts and towns with district status received funding, in 1995 it was 262, in the year 2000 354, and in 2016, programmes were deployed in 339 districts and towns with district status.

Distinguishing between the new federal states in eastern Germany and the old federal states in western Germany reveals that after 1994 considerably less regional development initiatives were deployed in the old federal states than in the new ones (Figure 6). The divide between East and West mentioned under Section 3.1 is thereby further accentuated.

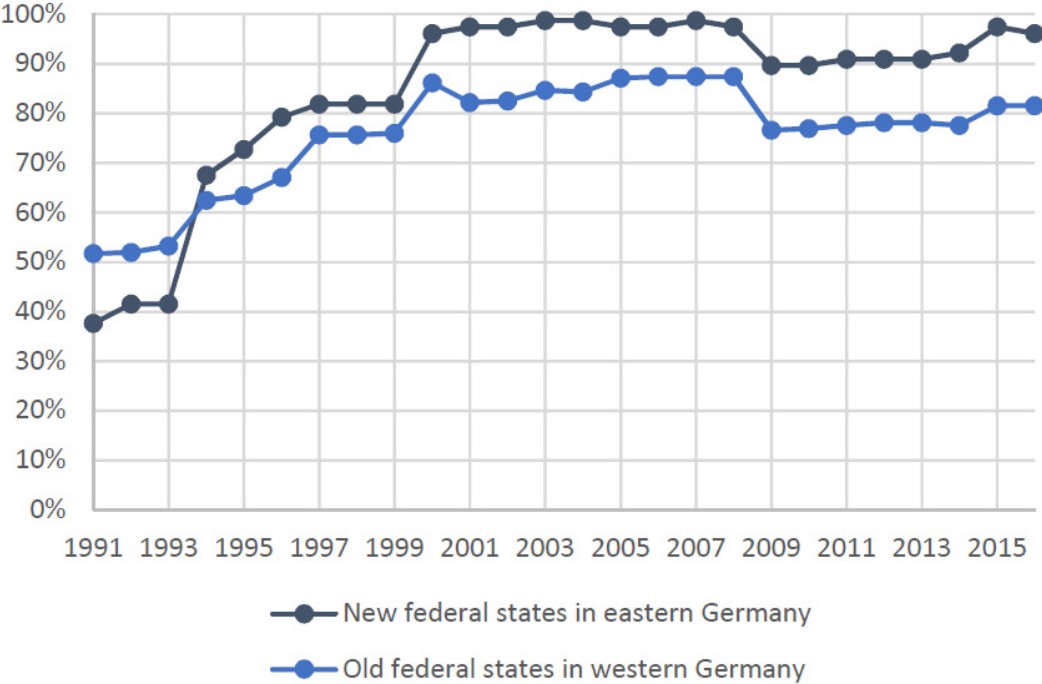

**Figure 6.** Deployment of programmes between 1991 and 2016 in the new federal states in eastern Germany and the old federal states in western Germany (relative figures). Source: authors' compilation (Department of Geography Giessen, Institute for Labour and the Economy Bremen).

Turning to the number of programmes deployed in districts categorised as 'sparsely populated rural districts' and 'rural districts with agglomeration tendencies', it is clear that, between 1991 and 2016, more initiatives in the analysed programmes were undertaken in sparsely populated districts than in rural districts with agglomeration tendencies (Figure 7). The temporal development of the funding can be summarised as being in accordance with the convergence objective, which is to concentrate support in a rather stable form on sparsely populated regions, particularly in eastern Germany. Paradigmatic discussions at the EU level (see Section 1) about focusing more on the stronger rural regions (rural districts with agglomerations tendencies) did not lead to a clearly visible change in priorities.

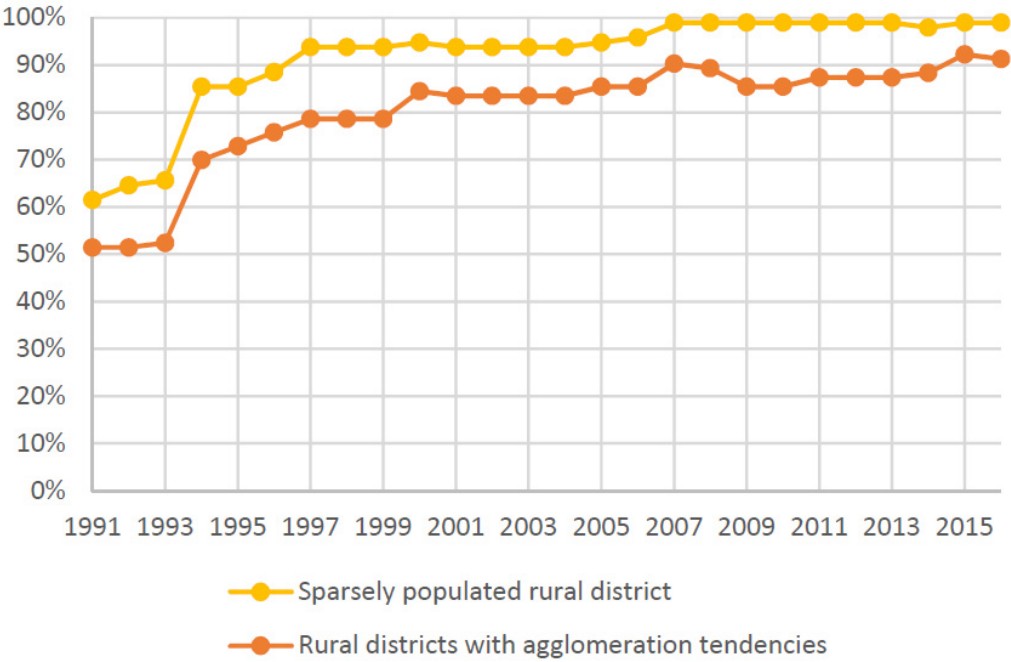

**Figure 7.** Deployment of programmes between 1991 and 2016 according to type of settlement structure in the districts (relative figures). Source: authors' compilation (Department of Geography Giessen, Institute for Labour and the Economy Bremen).

### 3.3. Programmes Impact

Attention is now directed towards the impacts of the programmes on the level of the districts and towns with district status. The number of programmes implemented in the districts and towns with district status was correlated with two indicators of regional development as dependent variables: population development rates and level and development rate of the regional GDP.

Figure 6 presents a comparison of the number of funding programmes in German districts and towns with district status and population trends. Across the board, the programmes aim to improve living conditions, so population trends are a more powerful indicator than economic development.

### 3.3.1. Population Trends

Between 1995 and 2014, the population decline was greater in the new federal states of eastern Germany (Ø = −12.52) than in the old federal states (Ø = −0.41). Examining correlations to the intensity of deployment of funding programmes shows no obvious influence on population trends between 1994 and 2014 in districts and towns with district status with a high to very high number of programmes. On the contrary, when compared to all districts and towns with district status (n = 402, Ø = −1.44), the districts with a high to very high number of funding programmes (n = 34) display relatively high population declines (Ø = −11.79). 195 of the German districts and towns with district status (ca. 48.5%), approximately half, experienced positive population trends in the period of analysis (see Table 2).

**Table 2.** Distribution of German districts and towns with district status according to population trends.

| Population Trends 1995–2014 | n = 402 | Examples |
|---|---|---|
| <−10% | 76 | Districts: Mecklenburgische Seenplatte, Holzminden, Birkenfeld |
| −10 to −5% | 50 | Districts: Regen, Märkischer Kreis, Oder-Spree |
| −5 to 0% | 81 | Districts: Rostock, Lippe, Sigmaringen |
| 0 to 5% | 92 | Districts: Fulda, Ostholstein, City: Leipzig |
| 5 to 10% | 62 | Districts: Emsland, Märkisch-Oderland, Lörrach |
| >10% | 41 | Districts: Oberhavel, Erding, Harburg |

Source: authors' compilation (Department of Geography Giessen, Institute for Labour and the Economy Bremen).

The population trends displayed in Figure 8 (left) only provide information on relative changes within a particular period of time. In order to ascertain how populated or under-populated a district or town with district status actually is, it is necessary to refer to actual population status, as displayed in Figure 8 (right) for 2014 (end of the period of analysis for population development).

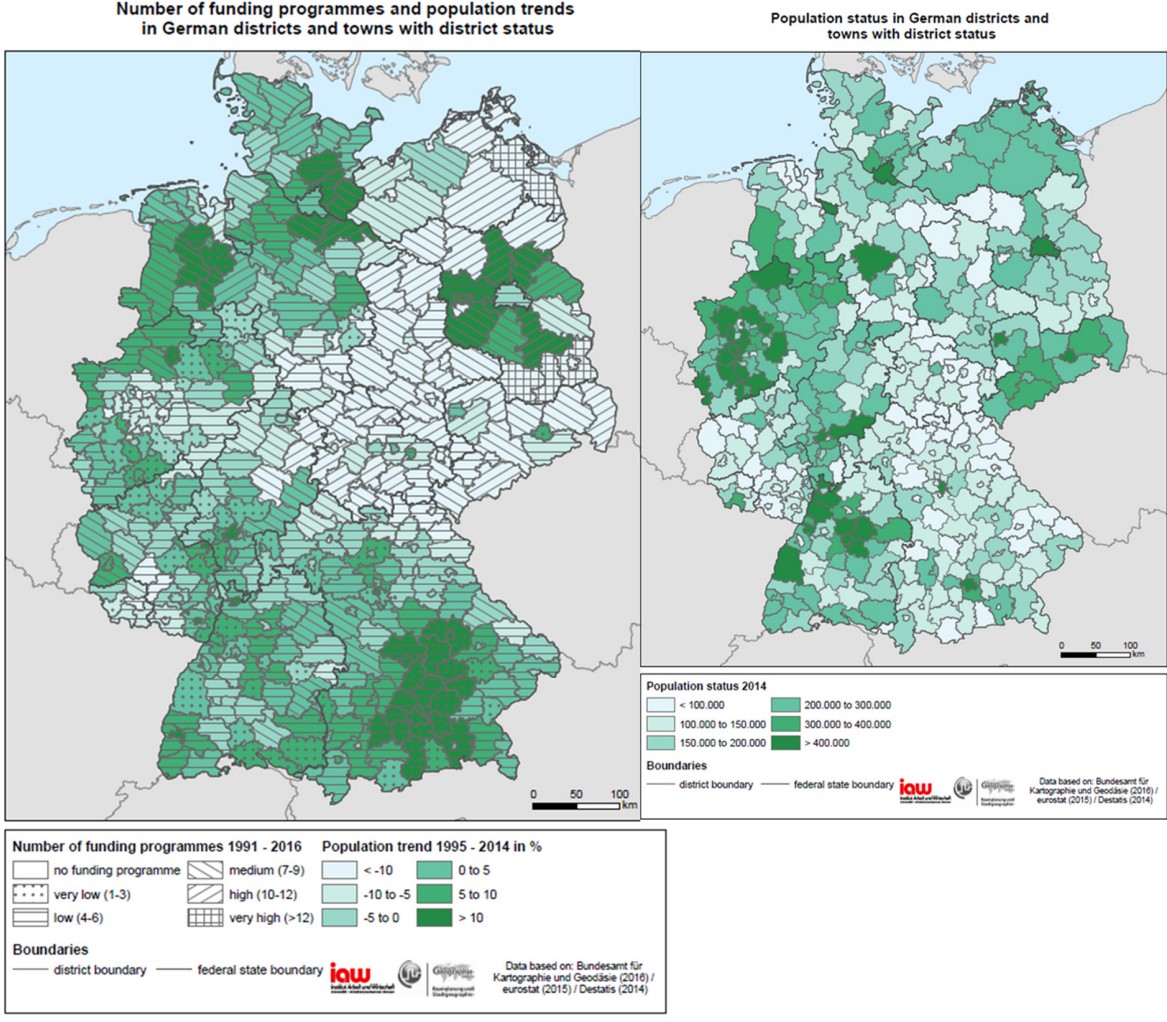

**Figure 8.** Number of programmes and population trends in German districts and towns with district status (**left**) and the population status of German districts and towns with district status (**right**). Source: authors' compilation (Department of Geography Giessen, Institute for Labour and the Economy Bremen).

3.3.2. Economic Strength

Figure 9 presents the number of funding programmes deployed in German districts and towns with district status in relation to economic strength. The period of analysis saw only positive growth rates of GDP, implying that GDP per capita increased throughout this time. In order to provide a context for the values, the countrywide growth rate (ca. 55%) is used for orientation, allowing the economic development of a district or city with district status to be classified as above or below average.

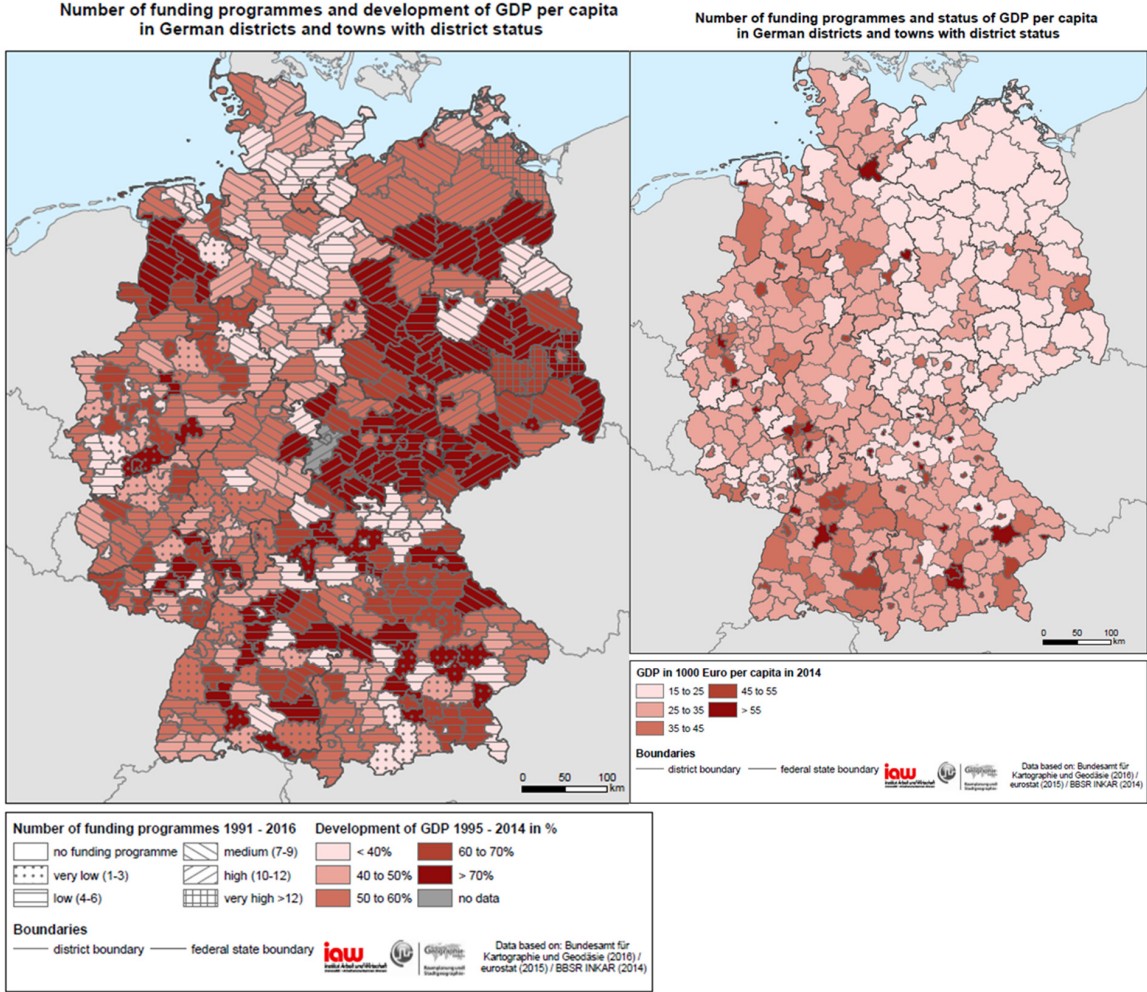

**Figure 9.** Number of programmes and development of gross domestic product (GDP) per capita in German districts and towns with district status (**left**) and status of GDP per capita in German districts and towns with district status (**right**). Source: authors' compilation (Department of Geography Giessen, Institute for Labour and the Economy Bremen).

In terms of the development of GDP per capita from 1995 to 2014, a slight East-West divide can be detected. The new federal states in eastern Germany and Bavaria experienced above-average increases in GDP per capita (Ø = 65.02). This is supported by considering the numbers or proportions of districts and towns with district status in the two halves of Germany outlined above. A total of 137 districts and towns with district status in the western half of Germany (ca. 61%) are characterised by increases in GDP that are below the national average. In contrast, in the new federal states and Bavaria (excluding the two territorial units for which it was impossible to calculate GDP trends due to a lack of data), there are only 65 districts and towns with district status (ca. 37%) with a growth rate of under 55%.

Despite this general difference, there are nonetheless contrary trends that may be identified. Thus, several districts in Brandenburg and Bayern show relatively low growth rates, while other districts

in the western half such as Unna, Leer or Heilbronn are characterised by considerably higher rates (Figure 8 and Table 3).

**Table 3.** Distribution of German districts and towns with district status according to development of GDP.

| GDP Growth Rates 1995–2014 | n = 402 | Examples |
|:---:|:---:|:---:|
| <40% | 77 | Districts: Barnim, Dithmarschen, Bayreuth |
| 40–50% | 80 | Districts: Vorpommern-Rügen, Vogelsbergkreis, Stade |
| 50–60% | 100 | Districts: Reutlingen, Nordfriesland, Meissen |
| 60–70% | 58 | Districts: Bautzen, Osnabrück, Traunstein |
| >70% | 85 | Districts: Rottweil, Uckermark, Vechta |
| No data | 2 | District: Wartburgkreis, City: Eisenach |

Source: authors' compilation (Department of Geography Giessen, Institute for Labour and the Economy Bremen).

Districts and towns with district status with a high to very high funding programme intensity (n = 34, over ten funding programmes) on average display a higher growth rate of GDP per capita (Ø = 67.96).

As already noted in the context of population trends, the development of GDP per capita displayed in Figure 8 (left) shows relative changes within a period of time. It is also the case here that, in order to draw conclusions about the actual economic strength of a district or town with district status (and also in comparison with other territorial units), the annual value for the end of the period of analysis must be utilised.

If GDP per capita in 2014 (as shown in Figure 8, right) is compared to the development of GDP from 1995 to 2014, there is no longer any sign of the East-West divide. It is rather the case that there is a divide between the old federal states (Ø = 35.4) and the new federal states (Ø = 25.3). Two conclusions can be drawn: firstly, in 2014 all the 402 districts and towns with district status in the old federal states were characterised by higher GDP per capita than the new federal states, although the cities with district status displayed notably higher averages throughout the country. Secondly, over the 20 years chosen for analysis of development trends it can be seen that while the new federal states in particular have comparatively low rates of GDP per capita, their long-term development tends to be more positive than that of the old federal states.

3.3.3. Statistical Analysis

The statistical analysis shows that a moderate negative significant correlation (R = −0.54) exists between the implementation of regional development funding and economic strength (average GDP per capita from the years 1995 and 2014) (see line one, column one in Table 4). The highest negative correlation can be found in the sparsely populated areas (see line five, column one in Table 4). This suggests that the programmes, in line with their intended purpose, are more intensively deployed in structurally weak regions. Furthermore, a correlation is found between the growth rate of GDP per capita between 1995 and 2014 and the number of programmes deployed. This is not, however, significant and is also much less pronounced (R = 0.12) (see line one, column two in Table 4. Looking at the types of regions (see lines two to five, column two in Table 4), the correlations disappear completely. The numbers in column three show that, between the types of "Rural districts with agglomeration tendencies" and "Sparsely populated rural districts", more regional cooperation was promoted in the regions with more inhabitants, which is not surprising.

A clearer picture is provided by the correlation between deployment of programmes and population trends. This is significantly negative (R = −0.38) (see line one, column four in Table 4), suggesting that the greater the number of programmes deployed in a district between 1995 and 2014,

the greater the decline in its population. This effect occurs significantly in all types of regions except for cities with district status (see column four, lines two to five in Table 4).

**Table 4.** Correlation between the number of funding programmes deployed and the economic growth/population trends of the German districts and towns with district status (1995–2014).

| | Districts/Towns with District Status | Pearson R-Correlation Coefficient | | | |
|---|---|---|---|---|---|
| | | Economic Development | | Demographic Development | |
| | | 1. Average GDP per capita 1995 and 2014 | 2. Development of GDP per capita 1995–2014 | 3. Average population status 1995 and 2014 | 4. Population trend 1995–2014 |
| 1. Total | 402 | −0.54 ** | 0.12 | −0.07 | −0.38 ** |
| 2. Cities with district status | 66 | −0.32 ** | 0.00 | 0.27 | −0.15 |
| 3. Urban districts | 137 | −0.36 ** | −0.01 | 0.08 | −0.35 ** |
| 4. Rural districts with agglomeration tendencies | 103 | −0.53 ** | −0.15 | 0.46 ** | −0.35 ** |
| 5. Sparsely populated rural districts | 96 | −0.63 ** | −0.3 ** | 0.48 ** | −0.33 ** |

Significance ** 0.5%. Source: authors' compilation (Department of Geography Giessen, Institute for Labour and the Economy Bremen). (Data from: BBSR ongoing spatial monitoring; DESTATIS).

Consideration of the individual types of settlement structure in the districts provides relatively clear findings. It can be generally concluded that the structurally weak regions are assisted more than structurally stronger regions. In terms of economic development and the deployment of programmes however, there is no significant indication that a high intensity of funding in the regions has a notable effect on regional development (measured by the two structural indicators).

The results related to population trends are confirmed when structurally similar sub-groups are analysed. With the exception of cities with district status, almost all the districts with other types of settlement structure are characterised by significant, moderate negative population development. It appears to be the case that the greater the deployment of funding programmes, the more negative the population trend. It should, however, be noted that population trends in particular are strongly influenced by factors that are not in the sphere of influence of the programmes considered here. Such factors include natural population development, employment opportunities and the housing market.

## 4. Discussion

In recent years, evaluation research has made considerable progress in terms of capturing the direct effects of funding programmes, particularly in the EU context. Most of them found that the funding had a positive impact on the economic development of the regions. However, the impacts were lower for the regions with the lowest economic performance, so that the political objectives of regional convergence were not reached sufficiently. Almost all evaluations were primarily aimed at assessing individual programmes. The few studies with a simultaneous examination of several programmes lead to different results: Some found differences between programmes concerning their impact, while few found positive relations between several programmes.

This study takes a further step in the direction of examining cumulative programme impacts. It is based on the analysis of 27 funding programmes, pilot projects and competitions from five policy fields, covering the period from 1991 to 2016. Its analyses are founded on the largest database of regional-development programmes implemented in Germany and the first attempt to detect cumulative effects of a large number of programmes over a long period. The most important findings can be summarised as follows.

- The programmes, competitions and pilot projects investigated are deployed in needy regions and take effect in a particular type of area that is targeted through the use of selective funding and territorial criteria: rural, structurally weak and disadvantaged. The concentration of programmes in various sub-areas and particular districts of Germany is, thus, to be expected.
- The project as a whole has also demonstrated that the deployment of programmes by the regions is very heterogeneous in terms of the timing and order of deployment, the running of programmes in parallel, and the actual content of the programmes. It can be seen that the regions strategically select the programmes they deploy and that they practise programme-hopping to achieve an almost seamless progression of programmes and ongoing funding over years. It has apparently been impossible to fulfil the ultimate objective of the programmes in the sense of creating and consolidating self-supporting cooperation structures and positive processes of regional economic development.
- The analysis provides no indication as to whether regions in which more initiatives were funded develop more positively than regions which received less assistance. No measurable cumulative effects on economic development could be demonstrated. The impact on population trend was shown to actually be negative. These findings support those of Panebianco [35] and also of Diller, Nischwitz and Kreutz [36]. This investigation is more informative than that published in 2014 as it considers a considerably greater number of programmes (27 instead of 18) and a longer period of time (going back to 1991).

The result seems rather surprising, considering the fact that the majority of other studies examining the effects of funding programmes have found significant dependencies between the level of funding and economic development indicators, even for individual programmes. We expected cumulative effects resulting from the combination of the different programmes. However, the GDP of the regions was not significantly influenced by the funding and the population development, even correlated negatively with the number of programmes implemented in the districts. This raises the question of what the possible reasons for the results of this study could be.

Firstly: an integrated regional development with its discursive bottom-up approach follows different principles to regional economic assistance: activating potentials is the primary objective [43] (p. 30). The majority of the programmes and initiatives considered here target 'soft' infrastructure and, in some cases, the regional networking of actors. The finance available is generally very limited in comparison to 'hard' programmes (e.g., EFRE) that fund regional economic development and large-scale infrastructure. For this reason, the measurable impacts could only be small. As the above discussion clarifies, the findings of the analysis do not mean that the deployed programmes and cooperation structures have had no impact. Di Cataldo and Monastiriotis [25] argue convincingly that measurable effects can generally be expected to be very low, and that the regional statistical indicators GDP and population trends cannot possibly capture the entire spectrum of the intended effects of regional initiatives. However, there is currently no alternative to using the few regional statistical indicators to attempt a medium to long-term analysis of the effects.

Secondly: even if the findings were not significant, the political consequences can be discussed. It is clearly necessary to pursue inquiries into the efficiency of the funding, investigating both the strategic framework levels and the regions. The practice debate is particularly concerned with the way in which funding is granted. However, is it reasonable to apply a distribution method that leads to a situation in which municipalities—especially in peripheral rural regions—are, on the one hand, unable to provide basic services and, on the other hand, consider it necessary to participate in complex competitions in order to acquire funding, sometimes involving complicated and communication-intensive projects? The discussion about the inefficiency of funding allocations and the problem of a lack of co-finance has repeatedly suggested the introduction of regional global budgets [28–30]. Despite a number of positive experiences, this approach has not been established in the EU or in Germany. It is, however, clearly time for a general scrutiny of the focus, structure and impact of the existing regional policy framework.

The third explanation seems even more relevant for the results: the limitations of this study. Instead of using the grant amounts as independent variables, we have simply counted the number of programmes implemented in each district. Even in this ambitious research project, it was not possible to completely capture and accurately allocate grant amounts with the given resources. However, we think that the mere number of programmes is a good indicator for the activity of the regions. In fact, some of the regions are very busy coupling several programmes, not only in terms of maximising the allocation of funds for the region, but also in order to achieve synergy effects. The low significant cumulative effects could therefore be interpreted speculatively as deadweight effects. However, to actually be able to draw such conclusions on a sound basis, an improved database is required. In this regard, an analysis that also takes into account the number of projects financed under the programmes, or even the volume of financing could lead to more profound or even contrary results. That leads to our conclusion.

## 5. Conclusions

Despite its limitations, the results of this study represent an important effort of regional research to investigate cumulative effects of a large number of overlapping regional development programmes. Its findings were presented and discussed on several political levels with actors from ministries, some regions (cases studies were as well a part of the research project, see Figure 1) and scientific institutions (e.g., Akademie für Raumentwicklung (ARL)) with interesting arguments for the further regional policy, e.g., aspects like regional budgets. However, the key implication of this part of the study, which attempts to measure the effects of regional funding, is primarily a methodological one for further research: the mere number of programmes supporting regional cooperation in one region gives a first indication to their activity, but is not sufficient for assessing the impacts of funding. In fact, the only information on the amount of regional subsidies will enable advanced analytical and statistical methods to be applied. Such an approach could provide more conclusive results and reveal more intervening variables than the rather simple descriptive analyses conducted in this project. The process of obtaining such detailed information on funding requires a great amount of effort. Many regional cooperation projects go beyond the boundaries of districts. However, districts are the fixed unit for the independent variables from the official statistic. This means: in order to determine the exact location of the funded projects, a spatial unit of investigation below the level of the districts is required: the municipalities. Expanding our existing database of almost 1500 regional cooperation projects demands a lot of resources. Given the fact that the digital facilities collecting such information have expanded in recent years, this challenge seems feasible.

**Author Contributions:** Conceptualization, original draft preparation, project Administration, C.D. and G.N.; writing review and editing, C.D.; methodology, validation, investigation, data curation, M.K. and P.C.; All authors have read and greed to the published version of the manuscript.

**Funding:** The project was founded by the Deutsche Forschungsgemeinschaft (DFG) (DI 1641/9-1).

**Conflicts of Interest:** None of the authors is involved in an interest conflict.

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
