# Peer review of "Regional Densities of Cooperation: Are There Measurable Effects on Regional Development?"

_urbansci, doi:10.3390/urbansci4030040_

Round 1

Reviewer 1 Report

This article aims at exploring a paradigm change in funding policies for rural regions towards a cooperative, actor-oriented regional development. The study investigates 27 funding programmes, pilot projects and competitions from five policy fields, covering the period from 1991 to 2016. The topic is very interesting and timely. Here are my comments to improve the paper:

1- Please break the current introduction section into two sections of introduction and literature review. Clearly list your research objectives in the introduction section.

2- the methodology section does not fully describe how you collected and analysed the data. Please provide much more details to help the readers understand what you have done.

3- the result section is rather descriptive. Try to be more analytical when discussing your findings.

4- the discussion and conclusion sections are rather short. Perhaps you need to write double than what you have now to do justice to your research and findings.

5- what are the limitations of your studies? You need to include these limitations somewhere in the paper.

6- what are the implications of your findings for the policy makers. In other words, how your research informs policy? You need one full paragraph on this.

7- what are your suggestions for future research? End the paper with this.

8- looking at your references, you have mostly looked at the literature in German language. Are there any relevant studies published in English? You have entirely overlooked these studies.

Author Response

See enclosed word-file.

Reviewer 2 Report

About the submission with the title "Regional densities of cooperation: are there measureable effects on regional development?" I have the following comments:

It could be important improve the literature review about the topic in the introduction section, considering other scientific productions from authors outside Germany.

The section 2 needs to be rewritten to describe in an understanding way for the readers. For example, what do the following sentences mean?

"In an initial step the programme regions corresponding to the programmes analysed were assigned to districts or towns with district status (Landkreis / kreisfreie Stadt) in Germany".

What is "districts or towns with district status"?

"The analysis evaluates whether or not a programme is deployed in a district or town with district status in Germany.".

What this explains?

"When selecting programmes for the investigation a medium scale was chosen. This refers to a spatial level above that of a municipality and below that of a federal state. There is a conscious emphasis
on programmes that focus on rural regions."

What this means? What is the scientific base for this?

"The number of programmes deployed in a district or town with district status was assessed. The utilisation of a programme was always only recorded once. If, for instance, several Local Action Groups (LAGs) of the LEADER programme existed in one district then they were only counted once so as to ensure comparability."

How you did that?

"Information about the programmes and their deployment in districts or towns with district status was captured in a broadly based, countrywide research process (internet, printed materials). A total of 1465 programme regions were entered into the database."

How you did this?

Can you put please the table 1 in an understanding way? And in English? Because I was unable to understand the explaination about this table. For example:

"The number of programmes deployed in the districts and towns with district status was correlated with two indicators of regional development as dependent variables."

Where you see this?

The remaing parts and section suffer from the same problem. Please, reanalyse the entire document.

The most serious aspect is about the scientific methodology considered. In fact, it is very descriptive and in this way the paper seems more a technical report. I suggest you consider something more robust for the data analyisis.

Author Response

See enclosed word file

Round 2

Reviewer 1 Report

Thanks for addressing the comments. I am happy to recommend the paper for publication. 

Author Response

Thank you for the constructive reviewing! We gave the manuscript to a native speaker for final corrections.

Reviewer 2 Report

I suggest the Authors avoid translate German text to English. 

For example table 1.

Author Response

Thank you for the constructive review. We gave the manuscript to a native language Speaker for final corrections.